# Knowledge, Attitudes, and Practices of Smallholder Dairy Cattle Farmers in Tanzania: A Cross-Sectional Survey on Cattle Infertility

**DOI:** 10.3390/vetsci12100993

**Published:** 2025-10-15

**Authors:** Athanas Ngou, Richard Laven, Timothy Parkinson, Isaac Kashoma, Daniel Donaghy

**Affiliations:** 1School of Veterinary Science, Massey University, Private Bag 11222, Palmerston North 4442, New Zealand; r.laven@massey.ac.nz (R.L.); t.j.parkinson@massey.ac.nz (T.P.); 2College of Veterinary Medicine and Biomedical Sciences, Sokoine University of Agriculture, Morogoro 67 115, Tanzania; kashoma@sua.ac.tz; 3School of Agriculture and Environment, Massey University, Private Bag 11222, Palmerston North 4442, New Zealand; d.j.donaghy@massey.ac.nz

**Keywords:** Tanzania, smallholder dairy farmers, dairy cattle infertility, knowledge, attitude, practices

## Abstract

Dairy cattle infertility is among the major challenges to smallholder dairy farming in Tanzania, yet it remains poorly understood by farmers. The current study investigated 301 farmers from the six key dairy-producing regions to explore their knowledge, attitudes and practices regarding dairy cattle infertility. Almost all farmers reported experiencing infertility in their herds and identified common infertility signs correctly, e.g., return to oestrus after breeding. Also, farmers reported poor feeding and housing, diseases and poor heat detection as important infertility causes, which were managed in various ways, including seeking assistance from veterinarians/livestock officers or culling. Our findings highlight that there is a wide recognition of dairy cattle infertility, but sometimes it is wrongly managed. Therefore, there is a need for the provision of better education to the farmers to reduce losses and improve herd productivity.

## 1. Introduction

Smallholder dairy cattle farming is an important component of the agricultural sector in many developing countries, particularly for rural and peri-urban households [1]. It plays a vital role in the improvement of the livelihoods of many households, as it ensures food security and enhances access to animal protein [2,3]. In Tanzania, smallholder cattle dairying contributes about 99% of the total milk produced [4], with ~97% being consumed at the household level or sold locally through informal value chains with little or no processing [5]. Of the total milk production, local cattle breeds contribute ~2.6 billion L/year (67%), with improved dairy cattle breeds producing ~1.3 billion L/year (33%) [6]. The improved dairy cattle are the crosses of European dairy breeds (e.g., Friesian, Ayrshire, and Jersey) with local Zebu, especially the Tanzanian Shorthorn Zebu, Boran, and Sahiwal. However, the demand for milk and milk products in Tanzania is ~12 billion L/year [6,7]. Consequently, ~75% of dairy consumption in Tanzania is of imported products. The growing demand in Tanzania for dairy products [8] means this is likely to increase unless the efficiency and productivity of dairy farming are improved [9].

Tanzania’s smallholder dairy farms produce most milk. However, Tanzanian smallholder dairy farmers face many constraints that limit their ability to improve the productivity of their farms. Most importantly, their farms are dominated by low-input-low-yield cattle breeds (~97% of cattle on their farms) [8], which limits their potential to increase yield. Other constraints include seasonal fluctuations in forage and feed availability, inability to access affordable veterinary and extension services, high levels of endemic disease and inadequate marketing of their products [8,10,11,12].

One key challenge facing smallholder dairy cattle farmers in Tanzania is the poor reproductive performance of their cattle [13,14,15]. This poor reproductive performance is characterised by low calving rates (65 births/100 cow years [13], prolonged intervals between calving and first observed oestrus (mean of 108 days [14]), and extended inter-calving intervals (mean 476–500 days [13,14]). This poor reproductive performance results in fewer calves being born, reduced lifetime milk production, and an increase in the proportion of cows in the herd that are not lactating. All these effects reduce household food security and limit the availability of milk for sale in the commercial market.

Improving reproductive performance on smallholder dairy farms will require significant input from farmers [16]. Obtaining such input will likely be dependent on farmer training and support programmes [14]. To optimise and target this training and support, data are needed on smallholder dairy cattle farmers’ current knowledge, attitudes and practices (KAPs) concerning dairy cattle infertility. However, no studies have been undertaken in Tanzania specifically focused on understanding these KAPs. Therefore, the purpose of this research is to address that knowledge gap (via evaluation of farmers’ KAPs on infertility) and provide insights to inform future efforts to improve fertility and increase the productivity of smallholder dairy farms in Tanzania. This research was conducted across six key dairying regions to capture regional variations in knowledge levels.

## 2. Methodology

This survey was undertaken alongside the survey described in a study by Ngou et al. [17]. A brief methodology is included here.

### 2.1. Ethical Considerations and Approval

This research was approved by the Ministry of Livestock and Fisheries through the Ethics Review Board of the Tanzania Livestock Research Institute (TALIRI) (reference number TLRI/RCC.21/007). The first author conducted interviews with the assistance of local veterinarians or livestock officers who introduced the Author to the farmer/respondent (usually the family head). Informed written consent was given before the interview. To ensure confidentiality, all responses were anonymised, and no identifying information was linked to the data.

### 2.2. Study Area and Study Farm Selection

This cross-sectional survey was undertaken between May 2022 and February 2023 in six dairy-producing regions of Tanzania (Figure 1). Within each region, convenience sampling was used to identify study villages and the first study farm in each village. Snowball sampling was then employed to select other study farms in that village. If the owner or someone who could respond to the questionnaire was not available on a selected farm, the interviewer moved to the next one on the list.

### 2.3. Field Data Collection

A questionnaire was developed to investigate a wide range of aspects of dairy-farming practice amongst smallholder dairy farmers. This paper mostly describes the responses to the questions related to the farmers’ KAPs around dairy cattle infertility. Pretesting of the questionnaire was undertaken with 24 smallholder farmers (not included in the final dataset) and 11 experts (veterinarians/livestock officers and researchers). KoboToolbox version 2025.2 (Cambridge, MA, USA) was used for offline data collection. The first section of this part of the questionnaire comprised enquiries on the signs indicating infertility and its associated causes. This section was included to ascertain whether farmers correctly understood the signs of infertility. This was followed by a section on farmers’ attitudes towards dairy cattle infertility on their farms. The last section asked the respondents about what they thought were their key practices on their farm concerning dairy cattle infertility, including its treatment, control, and prevention.

In the questionnaire, the term ‘repeat breeding’ was used to refer to cows that returned to oestrus after breeding, rather than the strict definition of the term that excludes cows with overt pathology of the reproductive system [18].

### 2.4. Data Management

Data from the questionnaire was downloaded from KoboToolbox to an Excel spreadsheet (Microsoft, Seattle, WA, USA). Data analysis was undertaken using SPSS version 25 (IBM, Seattle, WA, USA). Results for each question were tabulated and presented as overall results and by region. Where the effect of region on the answer to a question was thought to be of interest, logistic regression was used to analyse the impact of region upon the response, with the response to a question being the dependent variable and region the only predictor variable. Where responses in a category were <5% of respondents, categories were merged (based on proximity or compatibility) before analysis. Where this merging resulted in more than two ordered categories, ordinal logistic regression analysis was used, with the proportional odds assumption being tested using the test of parallel lines [19,20]. Multinomial logistic regression analysis was used, where the outcome was multinomial rather than ordinal (or the test of parallel lines had *p* < 0.05) [21].

For the section on the signs of infertility, each respondent was scored based on the number of correct answers. The effect of region on this score was analysed using an ordinal logistic regression with the respondent’s score as the outcome variable and region as the only predictor variable.

For all analyses, Tanga was used as the reference region. Descriptive data from Arusha (i.e., proportions of respondents) are reported, but data from Arusha were not included in any statistical analysis of the effect of the region due to the low number of respondents in that region.

## 3. Results

### 3.1. Signs of Infertility

Across all 301 respondents, 285 (95%) reported having experienced infertility in their dairy herds, a relatively uniform result across the study regions (range across regions: 94–98% of respondents; Figure 2 and Table 1). The high proportion of respondents reporting that they had experienced infertility was consistent with the high proportion who correctly understood the signs of infertility. Overall, the median number of questions on the signs of infertility that were answered correctly was 7/10 (ranging, at the individual farmer level, from 2 to 10 correct). Across the 10 separate questions, the proportion of correct respondents ranged from 282/301 (94%) who identified repeat breeding as a sign of infertility, to 199/301 (66%) who identified failure to produce a calf in a year.

Respondents from Morogoro had the highest average proportion of individual farmer correct answers (9/10), followed by those from Tanga and Kilimanjaro (8/10). In contrast, participants from Mbeya and Njombe had an average of 7 and 6 correct answers, respectively. Compared to Tanga, respondents in Morogoro had higher odds of having more questions correct (OR: 3.16, 95%CI: 1.59–6.30) while Njombe respondents had lower odds (OR: 0.21, 95%CI: 0.10–0.41).

To analyse the effect of region on the correct identification of individual signs of infertility the response ‘incorrect’ was combined with ‘not sure’ before analysis (i.e., both responses were defined as ‘not correct’), except for the two distractor questions (mastitis and poor milk production) where the answer ‘correct’ was combined with ‘not sure’ as ‘not correct’. Binary logistic regression analysis was then performed.

For the two distractor questions (i.e., producing less milk and suffering from mastitis), a high proportion of respondents correctly reported that *lower milk production* (83%) and *suffering from mastitis* (78%) were not signs of infertility. At the regional level, farmers from Morogoro were more likely to correctly identify less milk production or suffering from mastitis as not being signs of infertility than those from Tanga. The odds of a farmer from Morogoro identifying less milk production or suffering from mastitis as infertility signs were 0.06 (95%CI: 0.01–0.49) and 0.14 (95%CI: 0.03–0.66) times, respectively, those in Tanga.

*Repeat breeding* had the highest proportion of respondents identifying it as a sign of infertility (94%). Across the regions, this proportion varied from 100% (Arusha) to 82% (Njombe) (Table 1). The low proportion of incorrect respondents limited the power to identify the effect of region as, for example, compared to Tanga, respondents in Njombe had more than twice the odds of being ‘not correct’ for this question, but the wide 95% CI (OR: 2.18, 95%CI: 0.69–6.88) indicates that the data were compatible with both a moderate decrease and a large increase, hence no meaningful effect. Across all 301 respondents, 89% correctly identified abortion as a sign of infertility, ranging by region from 65% (Njombe) to 100% (Morogoro and Arusha). Farmers in Njombe had over 4 times greater odds of being ‘not correct’ for this question than farmers in Tanga (OR: 4.25, 95%CI: 1.54–11.8).

For *retained foetal membranes* (RFM), 83% of all 301 respondents correctly identified it as a sign of infertility; the range across regions was 57% (Njombe) to 97% (Kilimanjaro). As for abortion, respondents in Njombe had higher odds of being ‘not correct’ for this question (OR: 3.63, 95%CI: 1.48–8.90) than those in Tanga; and, in addition, farmers in Kilimanjaro had lower odds of being ‘not correct’ (OR: 0.15, 95%CI: 0.03–0.74) than those in Tanga. For *purulent vaginal discharge*, 81% of all respondents correctly identified it as a sign of infertility, ranging from 59% (Njombe) to 100% (Morogoro). Again, Njombe respondents had higher odds of being recorded as ‘not correct’ than those from Tanga (OR: 3.87, 95%CI: 1.53–9.78). Respondents from Mbeya (OR: 2.74, 95%CI 1.07–7.00) were also more likely to be ‘not correct’ than those from Tanga.

Regarding *reproductive diseases*, 79% of respondents correctly identified this as indicating infertility, with the highest proportion recorded in Arusha (100%) and the lowest in Mbeya (56%). As for purulent vaginal discharge, respondents from both Mbeya and Njombe had higher odds of being ‘not correct’ than respondents in Tanga (OR: 3.33, 95%CI: 1.39–7.95, and 2.53: 95%CI: 1.39–7.95, respectively). In contrast, respondents in the Morogoro region had lower odds of being ‘not correct’ (OR: 0.16, 95%CI: 0.03–0.75). For *stillbirths*, 78% of all respondents reported this correctly as an infertility sign, ranging from 98% (Morogoro) to 59% (Njombe). Once again, compared to those in Tanga, respondents from Njombe had higher odds (OR: 2.63, 95%CI: 1.11–6.19) of being ‘not correct’ and those in Morogoro had lower odds (OR: 0.07, 95%CI: 0.01–0.55).

For dystocia, 76% of all respondents were categorised as correct, with the highest proportion being recorded in Morogoro (95%) and the lowest in Njombe (59%). Compared to those from Tanga, respondents from Kilimanjaro had lower odds of being ‘not correct’ (OR: 0.25, 95%CI: 0.10–0.67), as did those from Morogoro (OR: 0.12, 95%CI: 0.03–0.43). As identified earlier, *failure to produce a calf per year* had the lowest proportion of correct respondents of all questions (66%). Across the regions, the largest proportion of ‘correct’ responses was in Tanga (83%) and the lowest in Morogoro (54%). Respondents from three regions had higher odds of being ‘not correct’ than those in Tanga: Morogoro (OR: 4.1, 95%CI: 1.69–9.95), Njombe (OR: 2.65, 95%CI: 1.07–6.59) and Kilimanjaro (OR: 2.44, 95%CI: 1.01–5.90).

For all 10 questions, the proportion of Njombe farmers who were correct was lower than in Tanga (Table 1), with the 95% CI for the OR excluding 1 for 6/10 questions. The equivalent figures for Mbeya, the region with the second highest proportion of ‘not correct’ answers, were that proportions correct were lower for 7/10 questions, and the 95%CI for the OR excluded 1 for 2/10 questions. In contrast, respondents in Kilimanjaro and Morogoro, the regions with the lowest proportion of ‘not correct’ answers across all 10 questions, had higher proportions of correct answers than those in Tanga for 8/10 and 9/10 questions, respectively, with the 95%CI for the OR excluding 1 for 2/10 and 5/10 questions.

### 3.2. Farmers’ Perceptions of the Causes of Cattle Infertility

The respondents’ understanding of the causes of infertility on their farms is summarised in Figure 3 and Table 2. To evaluate regional differences in farmers’ perceptions of the causes of infertility, three categories were created: (i) ‘agree’ (combining strongly agree and agree), (ii) ‘neutral’, and (iii) ‘disagree’ (combining disagree and strongly disagree). The effect of the region was analysed using ordinal logistic regression analysis, with ‘agree’ being higher than ‘neutral’, which was higher than ‘disagree’.

Most (93%) respondents agreed or strongly agreed that poor *nutrition and housing* caused infertility on their farm (feeding and housing were put together as adopted from a previous study [16] on smallholder dairy cow management and fertility). This was the highest proportion of ‘agree’ responses for any of the eight causes in this survey. At the regional level, the proportion of respondents agreeing ranged from 98% in Kilimanjaro and Morogoro to 84% in Njombe. As the proportion of neutral results overall (3%) was <5%, respondents in this category were merged with those in the category ‘disagree’ (to form a ‘neutral/disagree’ category) and a binomial logistic regression was used to compare the odds of a respondent in a region being ‘neutral/disagree’ rather than ‘agree’. However, although the fit of the model increased when the region was included (Akaike’s Information Criterion (AIC) was 24.7 compared to 32.5 for the intercept-only model), none of the individual region comparisons with Tanga had an OR whose 95%CI excluded 1 (Table 2). Most (89%) respondents agreed/strongly agreed that disease was a cause of infertility on their farm, ranging at the regional level from 96% (Morogoro) to 88% (Njombe). As only 2% of respondents disagreed that disease was a cause of infertility on their farm, this category was merged with neutral, and a binomial logistic regression was used to compare the odds of a respondent in a region being ‘disagree/neutral’ rather than ‘agree’. However, as for ‘poor nutrition and housing’, although the fit of the model slightly decreased when the region was included (AIC was 26.3 compared to 27.1 for the intercept-only model), none of the individual region comparisons with Tanga had an OR whose 95%CI excluded 1 (Table 2).

For *improper farm record keeping*, 85% of all the respondents agreed/strongly agreed that it was associated with infertility on their farm, ranging at the regional level from 97% (Kilimanjaro and Morogoro) to 60% (Mbeya). Respondents in both Mbeya and Njombe had higher odds of being in a lower category (i.e., were more likely to disagree than neutral or agree that improper record keeping is associated with infertility) than respondents in Tanga (OR: 6.16, 95%CI: 2.11–18.0 and OR: 3.36, 95%CI: 11.12–11.3, respectively). For *poor oestrus detection*, 83% of all the respondents agreed/strongly agreed that it caused infertility on their farm, ranging at the regional level from 98% (Kilimanjaro) to 66% (Mbeya and Njombe). Again, respondents from Mbeya and Njombe had higher odds (OR: 4.63, 95%CI: 1.67–12.9 and OR: 4.22, 95%CI: 1.51–11.8, respectively) of being in a lower category (e.g., were more likely to disagree than being neutral or agree) than those in Tanga. A small majority (58%) of all the respondents agreed/strongly agreed that *uterine infections* caused infertility on their farm, ranging from 89% (Morogoro) to 23% (Mbeya). Respondents’ odds of being in a lower category were higher in Mbeya (OR: 3.63, 95%CI: 1.65–7.96) and lower in Morogoro (OR: 0.11, 95%CI: 0.04–0.30) than respondents in Tanga.

The most common overall response (71%) of respondents concerning *embryo or foetal mortality* as a cause of infertility on their farm was ‘neutral’, ranging from 90% (Tanga) to 20% (Njombe). Only respondents in Njombe had lower odds (OR: 0.20, 95%CI: 0.08–0.48) of being in a lower category than respondents in Tanga. Similarly, 65% of all the respondents were neutral regarding the *improper use of hormones* as a cause of infertility, ranging from 92% (Kilimanjaro) to 20% (Njombe). Respondents in Mbeya had higher odds (OR: 5.36, 95%CI: 2.36–12.1) of being in a lower category than those in Tanga. Lastly, 78% of all the respondents’ responses regarding mycotoxins as an infertility cause were neutral, ranging from 90% (Kilimanjaro) to 53% (Njombe). Respondents in Mbeya had higher odds (OR: 8.24, 95%CI: 2.27–25.0), while those in Njombe had lower odds (OR: 0.19, 95%CI: 0.07–0.53) of being in a lower category than those in Tanga.

### 3.3. Farmers’ Attitudes Towards Infertility

A total of 285 respondents reported having encountered infertility challenges in their herds. Among those respondents, 98% (280/285) recognised that it was a problem for them, ranging from 100% (Tanga, Arusha, Kilimanjaro and Morogoro) to 92% (Njombe). For respondents who thought infertility was a problem in their herd, 48% (133/280) considered that it had major consequences, ranging from 58% (Morogoro) to 31% (Njombe) (Figure 4 and Table 3). The effect of region on the extent to which respondents thought infertility was a problem on their farm was analysed using multinomial logistic regression as the parallel lines assumption was not met (*p* < 0.05). Compared to the respondents in the Tanga region, only respondents in Njombe had higher odds of regarding infertility as a moderate (OR: 2.48, 95%CI: 1.05–5.86) or a minor (OR: 4.67, 95%CI: 1.04–20.8) problem on their farm rather than a major problem.

Three categories were created to evaluate regional differences in farmers’ attitudes regarding infertility: (i) ‘major’, (ii) ‘moderate’ and (iii) ‘negligible’ (merging of ‘minor’ and ‘not at all’). All analyses used ordinal logistic regression to assess the regional effect on farmers’ attitudes towards infertility issues on their farms, except for *repeat breeding*, *reproductive diseases*, and *stillbirth*, where binomial logistic regression was used and the categories used were ‘negligible’ against ‘other’ (merging of ‘major’ and ‘moderate’). Across the 285 respondents, 82% considered *repeat breeding* as a major infertility problem, ranging from 93% (Arusha) to 67% (Njombe). Compared to Tanga, only respondents in Njombe had lower odds (OR: 0.17, 95%CI: 0.05–0.56) of considering it a major issue rather than ‘moderate’ or ‘negligible’ using binomial logistic regression, i.e., major against other categories (merging of ‘moderate’ and ‘negligible’). For *failure to produce a calf in a year*, the most common response was moderate, with 44% of farmers considering it to have a moderate impact, ranging from 93% (Arusha) to 13% (Tanga). Compared to Tanga, respondents from Morogoro, Njombe, Kilimanjaro and Mbeya had higher odds of being in a lower category with OR (95%CI) of 38.0 (15.12–95.8), 23.0 (9.03–58.7), 7.64 (3.2–17.9), and 5.48 (2.2–13.7), respectively (Table 3).

The attitude of the majority of all the respondents (54%) was that the impact of *retained foetal membranes* (RFM) on infertility on their farm was negligible, ranging from 73% (Arusha) to 35% (Njombe). Respondents in Mbeya and Njombe had lower odds of being in the lower category (meaning that farmers in these regions were more likely to perceive RFM as a moderate or major issue on their farms) with OR (95%CI) of 0.44 (0.21–0.93) and 0.44 (0.21–0.91), respectively, compared to those in Tanga. *Dystocia* was considered to have a negligible impact as an infertility problem by 79% of all respondents, ranging from 93% (Arusha) to 63% (Njombe). There was no clear statistical difference between responses from Tanga compared to other regions, however, many of the 95%CI were wide (e.g., 0.98 to 5.62 for Njombe respondents) which means that, although our data were compatible with the absence of a regional effect, they were also compatible with a large effect of region (e.g., for Njombe respondents a large increase in their odds of being in a lower category) so we cannot exclude important differences between regions. The impact of *abortions* was considered to be similar to *dystocia*, with 80% of all respondents, ranging from 87% (Tanga and Arusha) to 75% (Mbeya), considering them to have a negligible impact. As for *dystocia*, no clear regional difference was identified in farmers’ attitudes regarding the impact of abortion, but again the 95%CI was wide (e.g., 0.16–1.20 for Mbeya vs. Tanga). An even higher proportion of respondents (93%) considered *stillbirths* to have a negligible impact on their farms. This ranged by region from 100% (Arusha) to 89% (Morogoro). Again, there were no clear statistical regional differences in farmers’ attitudes regarding the impact of *stillbirths*, but the 95% CI were wide (e.g., 0.21 to 5.55 for the Njombe region). Lastly, 93% of all respondents, ranging from 98% (Kilimanjaro and Tanga) to 60% (Njombe), considered reproductive disease to be a negligible problem on their farms. Respondents in Njombe had higher odds (OR: 9.95, 95%CI: 1.20–82.8) of considering reproductive diseases as having a negligible impact on their farms than respondents in Tanga.

### 3.4. Farmers’ Practices Regarding Dairy Cattle Infertility

Farmers’ practices were evaluated by allowing them to select from predefined options regarding the actions they take once they have an infertile animal on their farms (Table 4). To evaluate regional differences in farmers’ practices, three categories were made: (i) ‘agree’ (combining strongly agree and agree), (ii) ‘neutral’ and (iii) ‘disagree’ (merging disagree and strongly disagree), with that order being used for an ordinal regression.

Almost all farmers (94%), ranging from 100% (Tanga) to 88% (Njombe), reported *consulting a veterinary service provider* whenever they encountered an infertility case on their farms. Sixty-nine % of all respondents, ranging from 96% (Morogoro) to 48% (Tanga) agreed that *animal slaughter* was the best way to manage infertility. Compared to farmers in Tanga, farmers in Morogoro had higher odds of using slaughter to manage fertility (odds of being in a lower category (i.e., disagree (rather than agree/neutral) or disagree/neutral vs. agree) OR: 0.04, 95%CI: 0.01–0.17). Most farmers (58%), ranging from 80% (Arusha) to 52% (Mbeya), disagreed with the practice of *selling an infertile animal to another farmer*. No clear regional differences were identified concerning this practice. An even higher proportion of farmers (81%), ranging across the regions from 87% (Kilimanjaro) to 73% (Arusha and Njombe), disagreed with the practice of *self-treating an infertile animal* on their farm. However, no clear differences were noted across the region, but there was a wide 95% CI (Table 4).

In response to the question, “*Have you ever managed infertility using traditional methods?*” only eighteen farmers across the regions reported using traditional methods to address infertility. However, of those 18, 13 were in Njombe. Repeat breeding, RFM, and dystocia were among the conditions reported to be traditionally managed by smallholder dairy cattle farmers. One example of such is the extraction of the corner incisor teeth (Figure 5) as a “treatment” of a repeat breeder cow/heifer, which was practised by 13 farmers from the Njombe region. Other reported treatments included using natural herbs and receiving assistance from traditional healers.

## 4. Discussion

Infertility creates a substantial obstacle to improving the productivity and sustainability of smallholder dairy cattle farming in many developing countries, including Tanzania. The occurrence of infertility and other constraints to smallholder dairy farming arises from the combination of numerous factors, of which the KAPs of farmers are key [22,23]. Comprehensive knowledge of the KAPs of smallholder dairy cattle farmers concerning infertility is therefore a valuable start towards developing specific interventions to improve herd management, fertility and productivity (especially milk yield). Gathering these data requires acknowledging that smallholder dairy farming in Tanzania mostly operates under traditional practices, with limited access to modern veterinary services and education on fertility management.

Most of the farmers correctly understood what the signs of infertility were, with the majority correctly answering at least 7 of the questions (Table 1); however, there was significant farmer-to-farmer variation, with the lowest farmer only having 2/10 questions correct. Over 80% of the smallholder dairy cattle farmers across the regions were able to identify conditions like repeat breeding, abortion, retained foetal membranes (RFM) and purulent vulval discharge as indicators of infertility. However, a significant minority (34%) of farmers did not agree (or were uncertain) that an inability to produce a calf per year constituted infertility. This belief was highest among respondents from the Morogoro and Kilimanjaro regions, where farmers appear to prioritise milk yield and strategically manage calving intervals to meet specific farming or market objectives. Farmers can extend the lactation period by delaying insemination beyond 60–90 days postpartum, maximising milk production and potentially taking advantage of the higher milk prices during particular seasons. This intentional practice of producing calves less frequently than once yearly reflects a trade-off between reproductive efficiency and economic gains from milk production.

There was a clear effect of region on farmer recognition of the signs of infertility, with farmers from the Morogoro and Kilimanjaro regions appearing to be better informed than those from other regions, whilst Njombe farmers had the poorest levels of knowledge (Table 2). It is interesting to consider the extent to which this pattern reflects the education levels of the participants across regions; the majority of respondents from the Mbeya (66%) and Njombe (63%) regions had only primary education, while 68% and 46%, respectively, of farmers from Morogoro and Kilimanjaro had formal education beyond primary school [17]. This difference in education may be related to the Kilimanjaro region having the highest number of secondary schools of any of the regions in this study [24] and the location in Morogoro of Sokoine University of Agriculture (the only agricultural university in Tanzania). However, this study did not evaluate the association between the education level of the farmer and the availability of schools or universities. A previous study on the effect of education on production efficiency found a strong positive association in Asia and less in Latin America [25].

The causes of infertility that were highlighted by the majority of farmers as important on their farms were poor nutrition and housing, livestock disease, improper farm record keeping, poor oestrus detection and uterine infections, all of which have been highlighted as important by previous studies in both Tanzania [26] and in other similar systems outside of Tanzania, e.g., in Malawi: Banda, Kamwanja, Chagunda, Ashworth and Roberts [16]. The other infertility causes listed in Table 3 were seen as less important by our respondents. However, as the focus of this question was to ask the farmers their opinion on what was causing infertility on their farm, for most causes identified as less important, we do not know whether they were unimportant or whether they were important but were missed by our respondents. For example, embryo/foetal mortality can be difficult to recognise, especially when pregnancy diagnosis is uncommon, and can be a difficult concept for uneducated farmers to understand, so it may play a larger role in Tanzanian dairy infertility than suggested by this survey.

Most farmers across the regions regarded infertility as a significant (moderate to major) constraint on their farming; indeed, in most regions except for Njombe and Kilimanjaro, infertility was a major concern to the majority of farmers. Across all regions, repeat breeding was seen as an extremely important form of infertility. The perceived importance of repeat breeding is likely related to the ease of diagnosis (and its non-complex nature), with farmers directly observing cows repeatedly coming into oestrus without conceiving and recognising the instant economic losses from repeated breeding expenses.

There were significant differences across regions in opinions on the impact of failing to produce a calf annually. Across all respondents, 63% thought that failure to produce a calf annually had a moderate or smaller impact on farming. However, in the Tanga and Mbeya regions, failure to produce a calf annually was considered to have a major impact by 83% and 54% of respondents, respectively. This large difference across regions does not align with the underlying scientific consensus that the ideal calving interval for most cows is approximately 12 to 13 months [27], but rather likely reflects varying market conditions. In the Tanga and Mbeya regions, farmers’ cooperatives like CHAWAMU (Tanga) and UTAMBUZI (Mbeya) provide a year-round milk market, where there is limited seasonal variation in milk price. In this situation, it is clear that calving cows as frequently as feasible optimises milk production and milk income. Conversely, farmers in other regions (e.g., Morogoro) have highly seasonal pricing [28], with high prices and willing buyers in the dry season, compared to lower prices and production above demand during the rainy season [29] and thus smallholders can be tempted to alter breeding to target those prices. However, maximising milk production during the dry season poses greater challenges for farmers due to feed and water scarcity [28,30,31], which results in poorer fertility and a further increase in calving interval. Another study on beef reported that a calving date should consider not only the quantity but also the quality of the available feed [32].

Farmers’ perception of the importance of different infertility conditions is likely to depend on how often they observe these conditions. For example, most farmers would encounter cows that need multiple inseminations to become pregnant (i.e., returns to oestrus), so it is not surprising that they ascribe to it a high level of significance, although other forms of infertility (i.e., abortion, stillbirth, RFM) have a much greater impact on the productivity of the individual animal [33,34,35]. In other words, because of the lower frequency (compared to repeat breeding) of these latter forms of infertility, farmers tend to underestimate their impact. However, their occurrence not only impacts farmers in terms of costs associated with veterinary services but also affects the animals by reducing production [36,37] and shortening overall lifetime productivity [38,39]. For example, in Tanzania, a study by Kashoma and Ngou [40] showed that RFM causes a significant economic loss (i.e., 364,133.00 Tanzanian shillings (157.00 USD) per cow, reduced milk production (95.4%) and treatment costs (4.6%)). On top of these losses to the farmer, RFM leaves the animal sub-fertile even after successful treatment and recovery. Similarly, in developed countries, RFM has been reported to reduce milk sales and an overall reduction in animal fertility [41,42].

Farmers in this survey generally seemed to recognise the value of veterinary services, with 94% reporting that they would consult a veterinary service provider when facing any form of infertility on their farm, and 81% disagreeing with the practice of self-treating their infertile animals. Nonetheless, a minority of farmers continue to practice traditional methods of managing infertility, with 25% (13/49) of farmers from Njombe still extracting incisor teeth as a means of managing repeat breeding (Figure 5). This continued use of traditional methods suggests that there may be issues around the adequate provision of effective veterinary services, especially as the respondents knew that the first author (who was conducting the questionnaire) was a veterinarian and may have been reluctant to be negative about the role of veterinarians in managing infertility.

There was variation across respondents in how to manage persistently infertile animals. The majority (69%) supported the idea of culling infertile animals, consistent with infertility being an important cause of involuntary culling in East Africa [43]. However, 23% of all respondents supported selling infertile animals to other farmers. Respondents stated that they sold such animals to other regions or urban areas, particularly Dar es Salaam.

Furthermore, the main limitation of this study is its reliance on self-reported data; however, in the present study, we employed a guided interview as a strategy to overcome this limitation. We acknowledge that the use of selected regions in this study can limit the generalisation of our findings; nonetheless, these findings lay a good foundation for future research on infertility in smallholder dairy cattle farming in Tanzania.

## 5. Conclusions

The main objective of this study was to identify the knowledge gaps of smallholder dairy cattle farmers in Tanzania (i.e., via evaluation of farmers’ KAPs on infertility) and thus provide insights to improve fertility and productivity on such farms. There was marked variation in knowledge of the signs of infertility across regions, which may reflect the level of farmer education, but across regions, there was general agreement that repeat breeding was the most important infertility problem, and that nutrition/housing and disease were the main underlying causes of infertility. These results underscore the need for targeted, region-specific strategies to improve reproductive outcomes and highlight the importance of further qualitative studies to better understand regional differences. Furthermore, they also show that there is a need for improved education and outreach programmes to address farmers’ knowledge gaps and thereby enhance their capacity to manage and prevent infertility on their dairy farms.

## Figures and Tables

**Figure 1 vetsci-12-00993-f001:**
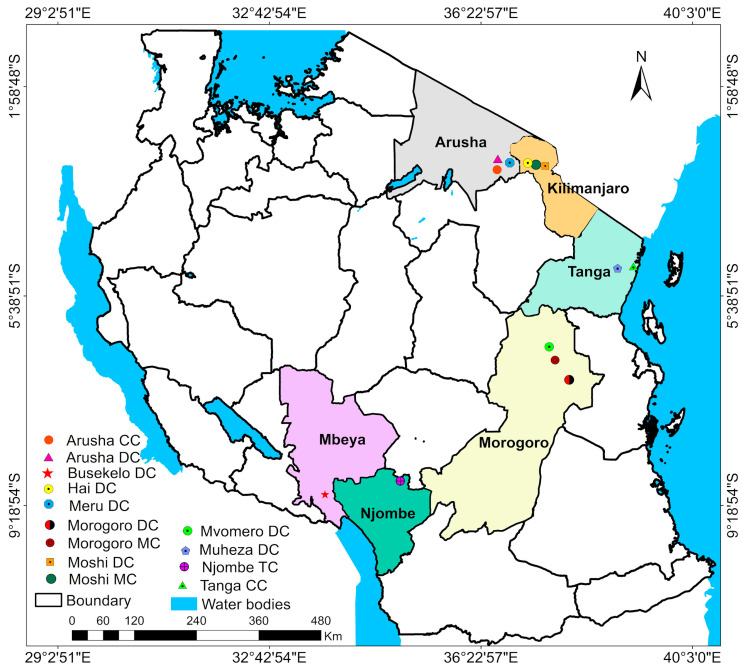
Map of Tanzania showing study regions and districts: Southern Highland regions, Northern Highland regions and Morogoro. NB: CC—City council, DC—District council and TC—Town council.

**Figure 2 vetsci-12-00993-f002:**
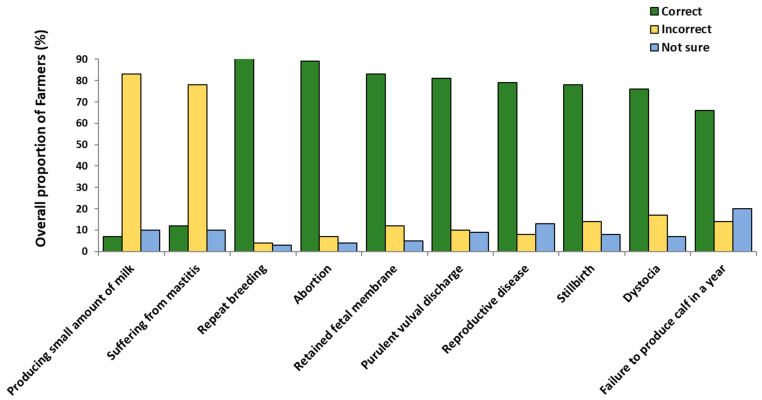
Smallholder dairy cattle farmers’ knowledge of the signs of infertility in dairy cattle in Tanzania mainland.

**Figure 3 vetsci-12-00993-f003:**
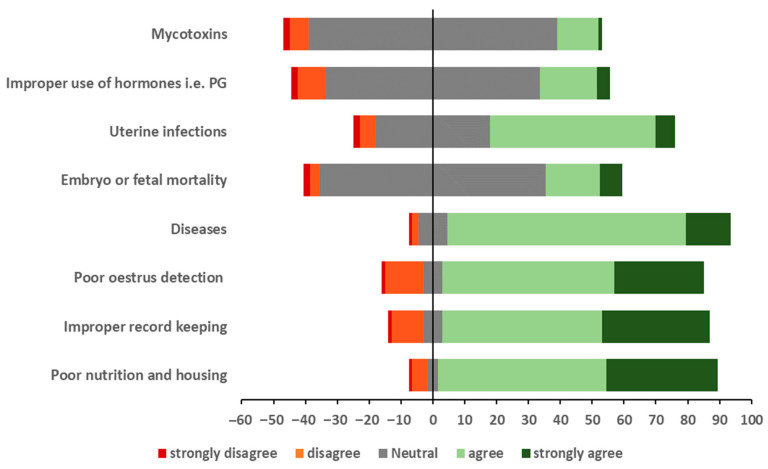
The opinions of Tanzanian smallholder dairy farmers on the causes of infertility on their farms. The x-axis is the overall percentage of respondents (n = 285) in each category, with ‘0’ being the midpoint of the ‘neutral’ response.

**Figure 4 vetsci-12-00993-f004:**
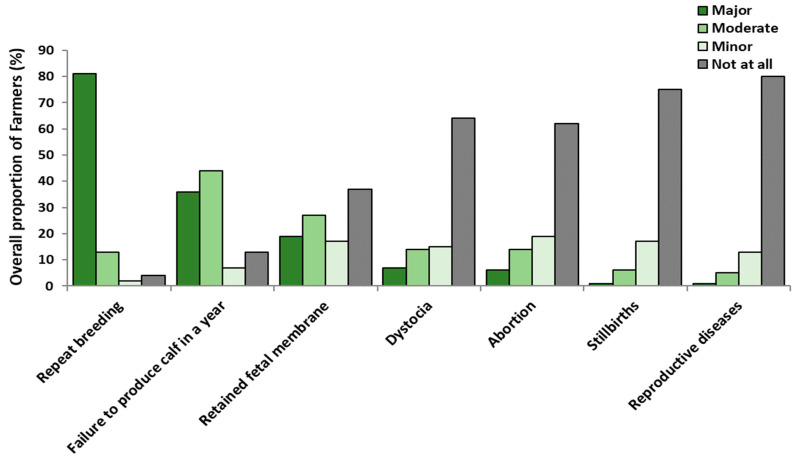
Perception of the Tanzanian smallholder dairy farmers (n = 285/301) on the impact of experienced infertility problems on their farm.

**Figure 5 vetsci-12-00993-f005:**
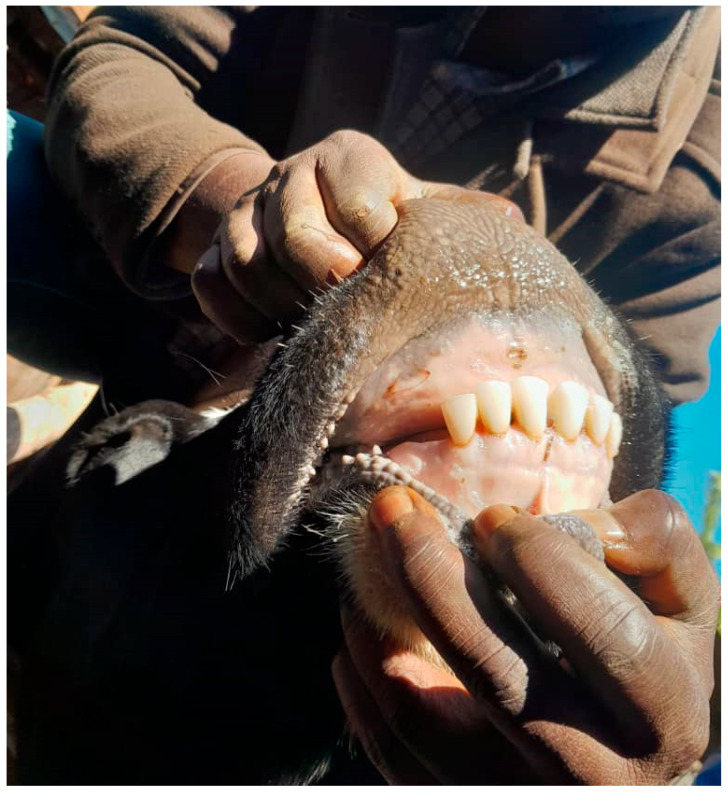
An animal without corner incisor teeth (removed as a management practice for repeat breeding) was observed in the Njombe region, Tanzania.

**Table 1 vetsci-12-00993-t001:** Opinions of Tanzanian smallholder dairy farmers on signs of infertility. Data are shown as the number of respondents (%).

Character	Category	Regions										Total
		Tanga	Arusha	Kilimanjaro		Mbeya		Morogoro		Njombe		n = 301 (%)
Total Respondents		n = 53(%)	n = 16(%)	n = 66(%)	OR (95%CI)	n = 55(%)	OR (95%CI)	n = 57(%)	OR (95%CI)	n = 54(%)	OR (95%CI)	
** *Have you ever encountered infertility on your farm?* **												
	yes	**52 (98)**	**15 (94)**	**62 (94)**	***	**52 (95)**		**55 (96)**		**49 (96)**		**285 (95)**
	No	1 (2)	1 (6)	2 (3)		3 (5)	***	1 (2)	***	4 (2)	***	12 (4)
I don’t know				2 (3)				1 (2)		1 (2)		4 (1)
**Sign of infertility**												
**Producing a small amount of milk**												
	correct	5 (9)		4 (6)	0.47 (0.18–1.26)	1 (2)	0.34 (0.11–1.05)		**0.06** **(0.01–0.49**)	11 (20)	2.17 (0.93–5.06)	21 (7)
	incorrect	**41 (77)**	**11 (69)**	**58 (88)**		**50 (91)**		**56 (98)**		**33 (61**)		**249 (83)**
	not sure	7 (13)	5 (31)	4 (6)		4 (7)		1 (2)		10 (19)		31 (10)
**Suffering from mastitis**												
	correct	6 (11)		12 (18)	1.78 (0.77–4.14)	7 (13)	0.96 (0.37–2.44)	1 (2)	**0.14** **(0.03–0.66)**	10 (19)	2.07 (0.87–4.84)	36 (12)
	incorrect	**42 (79)**	**14 (88)**	**45 (68)**		**44 (80)**		**55 (97)**		**35 (65)**		**235 (78)**
	not sure	5 (9)	2 (13)	9 (14)		4 (7)		1 (2)		9 (17)		30 (10)
**Repeat breeding**												
	correct	**48 (91)**	**16 (100)**	**65 (99)**	0.15 (0.02–1.31)	**53 (96)**	0.36 (0.07–1.96)	**56 (98)**	0.17 (0.02–1.52)	**44 (82)**	2.18 (0.69–6.88)	**282 (94)**
	incorrect	3 (6)				1 (2)				7 (13)		11 (4)
	not sure	2 (4)		1 (2)		1 (2)		1 (2)		3 (6)		8 (3)
**Abortion**												
	correct	**47 (89)**	**16 (100)**	**65 (99)**	0.12 (0.01–1.04)	**48 (87)**	1.14 (0.36–3.65)	**57 (100)**	***	**35 (65)**	**4.25** **(1.54–11.8)**	**268 (89)**
	incorrect	5 (9)		1 (1)		5 (9)				10 (19)		21 (7)
	not sure	1 (2)				2 (4)				9 (17)		12 (4)
**Retained foetal membrane (RFM)**												
	correct	**44 (83)**	**15 (94)**	**64 (97)**	**0.15** **(0.03–0.74)**	**42 (76)**	1.51 (0.59–3.91)	**54 (95)**	0.27 (0.07–1.07)	**31 (57)**	**3.63** **(1.48–8.90)**	**250 (83)**
	incorrect	7 (13)	1 (6)	1 (2)		9 (16)		3 (5)		15 (28)		36 (12)
	not sure	2 (4)		1 (2)		4 (7)				8 (15)		15 (5)
**Purulent vulval discharge**												
	correct	**45 (85)**	**15 (94)**	**58 (88)**	0.78 (0.27–2.23)	**37 (67)**	**2.74** **(1.07–7.00)**	**57 (100)**	***	**32 (59)**	**3.87** **(1.53–9.78)**	**244 (81)**
	incorrect	2 (4)	1 (6)	3 (5)		13 (24)				10 (19)		29 (10)
	not sure	6 (11)		5 (8)		5 (9)				12 (22)		28 (9)
**Suffering from reproductive disease**												
	correct	**43 (81)**	**16 (100)**	**60 (91)**	0.43 (0.15–1.27)	**31 (56)**	**3.33** **(1.39–7.95)**	**55 (97)**	**0.16** **(0.03–0.75)**	**34 (63)**	**2.53** **(1.05–6.11)**	**239 (79)**
	incorrect	1 (2)		1 (2)		14 (26)		1 (2)		7 (13)		24 (8)
	not sure	9 (17)		5 (8)		10 (18)		1 (2)		13 (24)		38 (13)
**Stillbirth**												
	correct	**42 (79)**	**15 (94)**	**55 (83)**	0.76 (0.30–1.93)	**35 (64)**	2.18 (0.92–5.17)	**56 (98)**	**0.07** **(0.01–0.55)**	**32 (59)**	**2.63** **(1.11–6.19)**	**235 (78)**
	incorrect	7 (13)	1 (6)	4 (6)		13 (24)		1 (2)		16 (30)		42 (14)
	not sure	4 (8)		7 (11)		7 (13)				6 (11)		24 (8)
**Dystocia**												
	correct	**36 (68)**	**12 (75)**	**59 (89)**	**0.25** **(0.10–0.67)**	**36 (66)**	1.12 (0.50–2.49)	**54 (95)**	**0.12** **(0.03–0.43)**	**32 (59)**	1.46 (0.66–3.21)	**229 (76)**
	incorrect	11 (21)	3 (19)	2 (3)		15 (27)		3 (5)		16 (30)		50 (17)
	not sure	6 (11)	1 (6)	5 (8)		4 (7)				6 (11)		22 (7)
**Failure to produce a calf in a year**												
	correct	**44 (83)**	7 (44)	**44 (67)**	**2.44** **(1.01–5.90)**	**38 (69)**	2.19 (0.87–5.47)	**31 (54)**	**4.1** **(1.69–9.95)**	**35 (65)**	**2.65** **(1.07–6.59)**	**199 (66)**
	incorrect	5 (9)	1 (6)	3 (5)		12 (22)		6 (11)		16 (30)		43 (14)
	not sure	4 (8)	**8 (50)**	19 (29)		5 (9)		20 (35)		3 (6)		59 (20)

For each region, the answer with the highest frequency (with its percentage in brackets) is in bold. OR: odds ratio; CI: confidence interval. ***: Indicates absence of figure for OR, 95%CI because of extremely low variabilities in farmers’ responses. Data from Arusha were not included in the analysis. Calculation and interpretation of the OR: For this analysis ‘incorrect’ and ‘not sure’ were merged as ‘not correct’ and compared to ‘correct’ using binomial logistic regression (except for mastitis/milk production where ‘correct’ and ‘not sure’ were merged as ‘not correct’ and compared to ‘incorrect). OR are odds, compared to Tanga, of a respondent from the region answering ‘not correct’; i.e., for RFM, the odds of a respondent from Njombe answering ‘not correct’ were 3.63 times higher than those for a respondent from Tanga. OR values in bold are those where 95% CI excludes 1.

**Table 2 vetsci-12-00993-t002:** Opinions of Tanzanian smallholder dairy farmers on the causes of infertility on their farms. Data are shown as the number of respondents (%).

Character	Category	Regions										Total
		Tanga	Arusha	Kilimanjaro		Mbeya		Morogoro		Njombe		
Total Respondents		n = 52 (18)	n = 15 (5)	n = 62 (22)	OR (95% CI)	n = 52 (18)	OR (95% CI)	n = 55 (19)	OR (95% CI)	n = 49 (17)	OR (95% CI)	n = 285(%)
**Poor nutrition and housing**												
Strongly agree		13 (25)	6 (40)	15 (24)	0.19 (0.02–1.82)	**32 (62)**	2.18 (0.61–7.75)	12 (22)	0.22 (0.02–2.06)	**26 (53)**	2.34 (0.66–8.34)	104 (37)
	Agree	**35 (67)**	**9 (60)**	**46 (74)**		12 (23)		**42 (76)**		15 (31)		**159 (56)**
	Neutral	2 (4)				3 (6)		1 (2)		1 (2)		7 (3)
	Disagree	1 (2)		1 (2)		5 (10)				7 (14)		14 (5)
Strongly disagree		1 (2)										1 (0.4)
**Diseases**												
Strongly agree		2 (4)		1 (2)	2.04 (0.59–7.05)	10 (19)	2.51 (0.72–8.75)		0.45 (0.08–2.58)	**25 (51)**	1.67 (0.44–6.33)	38 (13)
	Agree	**46 (88)**	**14 (93)**	**52 (84)**		**33 (63)**		**53 (96)**		18 (37)		**216 (76)**
	Neutral	4 (8)	1 (7)	9 (15)		5 (10)		2 (4)		4 (18)		25 (9)
	Disagree					4 (8)				2 (4)		6 (2)
Strongly disagree												
**Improper farm record-keeping**												
Strongly agree		14 (27)	7 (47)	13 (21)	0.30 (0.06–1.61)	5 (10)	**6.16** **(2.11–18.0)**	**34 (62)**	0.34 (0.06–1.83)	**23 (49)**	**3.76** **(1.12–11.3)**	96 (34)
	Agree	**33 (64)**	**8 (53)**	**47 (76)**		**26 (50)**		19 (35)		11 (22)		**144 (51)**
	Neutral			1 (2)		6 (12)		1 (2)		7 (14)		15 (5)
	Disagree	4 (8)		1 (2)		15 (29)		1 (2)		8 (16)		29 (10)
Strongly disagree		1 (2)										1 (0.4)
**Poor oestrus detection**												
Strongly agree		14 (27)	4 (27)	20 (32)	0.13 (0.02–1.13)	15 (29)	**4.63** **(1.67–12.9)**	9 (16)	0.78 (0.22–2.72)	**19 (39)**	**4.22** **(1.51–11.8)**	81 (28)
	Agree	**32 (62)**	**10 (67)**	**41 (66)**		**19 (37)**		**41 (75)**		13 (27)		**156 (55)**
	Neutral	4 (8)	1 (7)			2 (4)		3 (6)		5 (10)		15 (5)
	Disagree	2 (4)		1 (2)		16 (31)		2 (4)		12 (25)		33 (12)
Strongly disagree												
**Uterine infections**												
Strongly agree				1 (2)	0.54 (0.26–1.11)	1 (2)	**3.63** **(1.65–7.96)**		**0.11** **(0.04–0.30)**	15 (31)	0.59 (0.27–1.29)	17 (6)
	Agree	23 (44)	**12 (80)**	**37 (60)**		11 (21)		**49 (89)**		**16 (33)**		**148 (52)**
	Neutral	**29 (56)**	3 (20)	24 (39)		**29 (56)**		6 (11)		13 (27)		104 (37)
	Disagree					11 (21)				2 (4)		13 (5)
Strongly disagree										3 (6)		3 (1)
**Embryo or foetal mortality**												
Strongly agree		2 (4)			0.80 (0.35–1.82)	5 (10)	0.66 (0.28–1.57)		0.67 (0.29–1.56)	13 (27)	**0.20** **(0.08–0.48)**	20 (7)
	Agree	3 (6)	2 (13)	9 (15)		8 (15)		10 (18)		**16 (33)**		48 (17)
	Neutral	**47 (90)**	**13 (87)**	**53 (85)**		**35 (67)**		**45 (82)**		10 (20)		**203 (71)**
	Disagree					4 (8)				7 (14)		11 (4)
Strongly disagree										3 (6)		3 (1)
**Improper use of hormones, i.e., PGF2α**												
Strongly agree		1 (2)	1 (7)		1.59 (0.76–3.34)		**5.36** **(2.36–12.1)**	1 (2)	1.13 (0.53–2.43)	5 (10)	1.11 (0.46–2.67)	8 (3)
	Agree	12 (23)	2 (13)	4 (7)		4 (8)		9 (16)		**17 (35)**		48 (17)
	Neutral	**35 (67)**	**12 (80)**	**57 (92)**		**29 (56)**		**43 (78)**		10 (20)		**186 (65)**
	Disagree	3 (6)		1 (2)		18 (35)		2 (4)		2 (4)		26 (9)
Strongly disagree		1 (2)				1 (2)				15 (31)		17 (6)
**Mycotoxins**												
Strongly agree					0.66 (0.23–1.88)		**8.24** **(2.27–25.0)**		0.48 (0.17–1.38)	3 (6)	**0.19** **(0.07–0.53)**	3 (1)
	Agree	5 (10)	2 (13)	6 (10)				8 (15)		16 (33)		37 (13)
	Neutral	**44 (85)**	**13 (87)**	**56 (90)**		**37 (71)**		**47 (86)**		**26 (53)**		**223 (78)**
	Disagree					14 (27)				3 (6)		17 (6)
Strongly disagree		3 (6)				1 (2)				1 (2)		5 (2)

**NB:** The answer with the highest frequency (with its percentage in brackets) for each region is bolded. OR: odds ratio; CI: confidence interval. Data from Arusha were not included in the analysis. Calculation and interpretation of the OR: For this analysis, the effect of region on the proportion of respondents in the categories ‘agree’ (combining strongly agree and agree), ‘neutral’ and ‘disagree’ (combining disagree and strongly disagree) were assessed using ordinal logistic regression, except for ‘poor nutrition and housing’ and ‘livestock diseases’ where binomial logistic regression was used (i.e., agree against neutral/disagree) because their overall responses for neutral or disagree category was <5%). OR are odds, compared to Tanga, of a respondent being in a lower category, e.g., regarding whether mycotoxins were important, Njombe respondents had lower odds (OR: 0.19. 95%CI: 0.07–0.53) of being in a lower category (e.g., disagree rather than neutral/agree) than farmers in Tanga, i.e., Njombe farmers were more likely to think mycotoxins were important than farmers in Tanga. OR values in bold are those where the 95%CI excludes 1.

**Table 3 vetsci-12-00993-t003:** Perceived importance of animal infertility cases experienced by smallholder dairy cattle farmers in Tanzania. Data are shown as the number of respondents (%).

Character	Category	Regions										Total
		Tanga	Arusha	Kilimanjaro		Mbeya		Morogoro		Njombe		
Total Respondents		n = 52 (18)	n = 15 (5)	n = 62 (22)	OR (95% CI)	n = 52 (18)	OR (95% CI)	n = 55 (19)	OR (95% CI)	n = 49 (17)	OR (95% CI)	n = 285(%)
**Do you consider infertility a problem on your farm?**												
**Total respondents**		**n = 52 (19)**	**n = 15 (5)**	**n = 62 (22)**	**OR (95% CI)**	**n = 51 (18)**	**OR (95% CI)**	**n = 55 (20)**	**OR (95% CI)**	**n = 45 (16)**	**OR (95% CI)**	**n = 285(%)**
	Yes	**52 (100)**	**15 (100)**	**62 (100)**	***	**51 (98)**	***	**55 (100)**	***	**45 (92)**	***	**280 (98)**
	No					1 (2)				4 (8)		5 (2)
**(If yes), To what extent is infertility a problem on your farm?**												
**Total respondents**		**n = 52 (19)**	**n = 15 (5)**	**n = 62 (22)**	**OR (95% CI)**	**n = 51 (18)**	**OR (95% CI)**	**n = 55 (20)**	**OR (95% CI)**	**n = 45 (16)**	**OR (95% CI)**	**n = 280(%)**
	Major	**28 (54)**	**8 (53)**	25 (40)		**26 (51)**		**32 (58)**		14 (31)		**133 (48)**
	Moderate	21 (40)	6 (40)	**33 (53)**	1.76 (0.82–3.79)	22 (43)	1.13 (0.51–2.52)	17 (31)	0.71 (0.31–1.60)	**24 (53)**	**2.48** **(1.05–5.86)**	123 (44)
	Minor	3 (6)	1 (7)	4 (6)	1.49 (0.30–7.33)	3 (6)	1.08 (0.20–5.82)	6 (11)	1.75 (0.40–7.66)	7 (16)	**4.67** **(1.04–20.8)**	24 (9)
**What is the importance of these infertility problems on your farm?**												
**Repeat breeding ^(b)^**												
	Major	**48 (92)**	**14 (93)**	**50 (81)**	0.35 (0.11–1.15)	**42 (81)**	0.35 (0.10–1.20)	**47 (85)**	0.49 (1.14–1.74)	**33 (67)**	**0.17** **(0.05–0.56)**	**234 (82)**
	Moderate	3 (6)	1 (6)	9 (15)		7 (13)		5 (9)		11 (22)		36 (13)
	Minor			1 (2)		1 (2)		2 (4)		1 (2)		5 (2)
	Not at all	1 (2)		2 (3)		2 (4)		1 (2)		4 (8)		10 (4)
**Failure to produce a calf in a year ^(a)^**												
	Major	**43 (83)**	1 (7)	19 (31)	**7.64** **(3.2–17.92)**	**28 (54)**	**5.48** **(2.20–13.7)**	3 (5)	**38.0** **(15.1–95.8)**	10 (20)	**23.0** **(9.03–58.7)**	103 (36)
	Moderate	7 (13)	**14 (93)**	**41 (66)**		12 (23)		**30 (55)**		**22 (45)**		**126 (44)**
	Minor	1 (2)		2 (3)		3 (6)		12 (22)		2 (4)		20 (7)
	Not at all	1 (2)				9 (17)		10 (18)		12 (31)		36 (12)
**Retained foetal membrane ^(a)^**												
	Major	7 (13)	3 (22)	7 (11)	1.39 (0.66–2.92)	**17 (33)**	**0.44** **(0.21–0.93)**	7 (13)	1.09 (0.52–2.30)	12 (24)	**0.44** **(0.21–0.91)**	53 (19)
	Moderate	15 (29)	1 (7)	14 (23)		13 (25)		15 (27)		**20 (41)**		78 (27)
	Minor	10 (19)	4 (27)	13 (21)		8 (15)		9 (16)		4 (8)		48 (17)
	Not at all	**20 (38)**	**7 (47)**	**28 (45)**		14 (27)		**24 (44)**		13 (27)		**106 (37)**
**Dystocia ^(a)^**												
	Major	2 (4)	1 (7)	1 (2)	0.41 (0.14–1.19)	6 (12)	1.35 (0.54–3.34)	2 (4)	0.84 (0.33–2.16)	8 (16)	2.35 (0.98–5.62)	20 (7)
	Moderate	9 (17)		5 (8)		7 (13)		8 (15)		10 (20)		39 (14)
	Minor	12 (23)	3 (20)	5 (8)		12 (23)		8 (15)		3 (6)		43 (15)
	Not at all	**29 (56)**	**11 (73)**	**51 (82)**		**27 (52)**		**37 (67)**		**28 (57)**		**183 (64)**
**Abortion ^(a)^**												
	Major	1 (2)	1 (7)	3 (5)	0.54 (0.20–1.44)	6 (12)	0.43 (0.16–1.20)	5 (9)	0.65 (0.23–1.87)	1 (2)	0.62 (0.22–1.78)	17 (6)
	Moderate	6 (12)	1 (7)	11 (18)		7 (13)		5 (9)		9 (18)		39 (14)
	Minor	13 (25)	2 (13)	12 (19)		6 (12)		18 (33)		2 (4)		53 (19)
	Not at all	**32 (62)**	**11 (73)**	**36 (58)**		**33 (63)**		**27 (49)**		**37 (76)**		**176 (62)**
**Stillbirths ^(b)^**												
	Major	1 (2)			1.13 (0.24–5.28)	1 (2)	1.74 (0.39–7.68)	1 (2)	2.00 (0.47–8.45)		1.07 (0.21–5.55)	3 (1)
	Moderate	2 (4)		4 (6)		4 (8)		5 (9)		3 (6)		18 (6)
	Minor	13 (25)	4 (27)	9 (15)		7 (13)		12 (22)		4 (8)		49 (17)
	Not at all	**36 (69)**	**11 (73)**	**49 (79)**		**40 (77)**		**37 (67)**		**42 (86)**		**215 (75)**
**Reproductive diseases ^(b)^**												
	Major	1 (2)			0.84 (0.05–13.7)		2.04 (0.18–23.2)	1 (2)	5.10 (0.58–45.2)	2 (4)	**9.95** **(1.20–82.8)**	4 (1)
	Moderate		2 (13)	1 (2)		2 (4)		4 (7)		6 (12)		15 (5)
	Minor	6 (12)	1 (7)	7 (11)		4 (8)		17 (31)		3 (6)		38 (13)
	Not at all	**45 (87)**	**12 (80)**	**54 (87)**		**46 (88)**		**33 (60)**		**38 (78)**		**228 (80)**

**NB:** The answer with the highest frequency (with its percentage in brackets) for each region is bolded. OR: odds ratio; CI: confidence interval. ***: Indicates absence of figure for OR, 95%CI because of lack of or extremely low variabilities in farmers’ responses. Data from Arusha were not included in the analysis. Calculation and interpretation of the OR: For this analysis, i.e., the effect of region on the extent of experienced infertility cases on farms by smallholder dairy cattle farmers, three categories were created: ‘major’, ‘moderate and ‘negligible’ (merging of ‘minor’ and ‘not at all’). All analyses used ordinal logistic regression ^(a)^, except for repeat breeding, reproductive diseases and stillbirth where binomial logistic regression ^(b)^ was used and the categories used were ‘negligible’ against ‘other’ (merging of ‘major’ and ‘moderate’), because one of the categories had less than 5% proportion of the total, i.e., only 1% of the respondents considered reproductive diseases to have major impact on their farm. OR are odds, compared to Tanga, i.e., Njombe respondents had higher odds (OR: 9.95, 95%CI: 1.20–82.8) of regarding stillbirths as having a negligible impact on their farms compared to those in Tanga and relative to ‘other’ (major or moderate). OR values in bold are those where 95%CI excludes 1.

**Table 4 vetsci-12-00993-t004:** Practices of the smallholder dairy cattle farmers in Tanzania when they encounter infertile cattle on their farm. Data are shown as the number of respondents (%).

Character	Category	Regions										Total
		Tanga	Arusha	Kilimanjaro		Mbeya		Morogoro		Njombe		
Total Respondents		n = 52 (18)	n = 15 (5)	n = 62 (22)	OR (95% CI)	n = 52 (18)	OR (95% CI)	n = 55 (19)	OR (95% CI)	n = 49 (17)	OR (95% CI)	n = 285(%)
** *What do you do when you have an infertile cow on your farm?* **												
**Report to the veterinary service provider**												
Strongly agree		20 (38)	4 (27)	12 (19)	***	**28 (54)**		20 (36)	***	**37 (71)**		121 (42)
Agree		**32 (62)**	**10 (67)**	**48 (77)**		19 (37)		**31 (56)**		9 (17)		**149 (52)**
Neutral			1 (7)	1 (2)		1 (2)	***	4 (7)		2 (4)	***	9 (3)
Disagree				1 (2)		4 (8)				3 (6)		8 (3)
Strongly disagree										1 (2)		1 (0.3)
**Slaughter the animal**												
Strongly agree		4 (8)	4 (27)	13 (21)	0.16 (0.07–0.39)	4 (8)	0.98 (0.47–2.03)	15 (27)	**0.04** **(0.01–0.17)**	**16 (31)**	0.87 (0.41–1.84)	56 (19)
Agree		**21 (40)**	**10 (67)**	**40 (65)**		**23 (44)**		**38 (69)**		13 (25)		**145 (50)**
Neutral		10 (19)	1 (7)	6 (10)		6 (12)				5 (10)		28 (10)
Disagree		16 (31)		3 (5)		19 (37)		2 (4)		17 (33)		57 (20)
Strongly disagree		1 (2)								1 (2)		2 (1)
**Sell to other farmers**												
Strongly agree					0.72 (0.35–1.48)	1 (2)	0.56 (0.26–1.21)		0.80 (0.38–1.67)	3 (6)	0.60 (0.24–1.50)	4 (1)
Agree		10 (19)	3 (20)	13 (21)		16 (31)		9 (16)		11 (21)		62 (22)
Neutral		9 (17)		16 (26)		8 (15)		16 (29)		5 (10)		54 (19)
Disagree		**30 (58)**	**12 (80)**	**33 (53)**		**27 (52)**		**29 (53)**		**23 (44)**		**154 (53)**
Strongly disagree		3 (6)						1 (2)		10 (19)		14 (5)
**Treat the animal by myself**												
Strongly agree			1 (7)	1 (2)		2 (4)		1 (2)		2 (4)		7 (2)
Agree		6 (12)	3 (20)	5 (8)	1.22 (0.43–3.51)	6 (12)	0.85 (0.30–2.40)	8 (15)	0.80 (0.29–2.20)	11 (21)	0.45 (0.17–1.15)	39 (14)
Neutral		2 (4)		2 (3)		1 (2)		1 (2)		1 (2)		7 (2)
Disagree		**37 (71)**	**11 (73)**	**52 (84)**		**41 (79)**		**38 (69)**		**29 (56)**		**208 (72)**
Strongly disagree		7 (13)		2 (3)		2 (4)		7 (13)		9 (17)		27 (9)

**NB:** The answer with the highest frequency (with its percentage in brackets) for each region is bolded. OR: odds ratio; CI: confidence interval. Data from Arusha were not included in the analysis. ***: Indicates absence of figure for OR, 95%CI because of lack of or extremely low variabilities in farmers’ responses. Calculation and interpretation of the OR: For this analysis, three categories were made: ‘agree’ (combining strongly agree and agree), ‘neutral’ and ‘disagree’ (combining disagree and strongly disagree). Analyses were performed using ordinal logistic regression with agree being the highest category and disagree the lowest. OR are odds ratios compared to Tanga, e.g., Morogoro respondents had lower odds (OR: 0.04, 95%CI: 0.01–0.17) of being in a lower category than respondents in Tanga, regarding the use of animal slaughter to manage infertility. OR values in bold are those where 95%CI excludes 1.

## Data Availability

The generated and analysed datasets used in this study are available from the corresponding author upon request.

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
