# Peer review of "Knowledge, Attitudes, and Practices of Smallholder Dairy Cattle Farmers in Tanzania: A Cross-Sectional Survey on Cattle Infertility"

_vetsci, 2025, doi:10.3390/vetsci12100993_

Round 1

Reviewer 1 Report

Comments and Suggestions for Authors

I thank the authors for presenting this work. 
The topic addressed I believe is of substantial importance. Understanding the knowledge baseline of a production system, in this case that of a developing apese. defining the potential and limitations is fundamental to the evolution of an industry. 
However, the work needs revisions that will gliin its scientific value and comprehensibility and allow a better understanding of the results and state of the art generated.

Line 14: little is known about what?

Introduction

I suggest that the introduction be improved by also reporting the breeds that are majorly bred in order to describe what is the livestock stock but also the production potential of these breeds.

Methodology

It may be helpful to report the requested data and specify the mean of the questions. There are the range for example for milk productivity to define low or high?

Results

I suggest to add a descriptive statistics of the farms involved into the study (e.g, herd composition, numbers of animals for physiological category, milk yield, age of the farmers and level of edeucation…)

I suggest formatting the tables to make them more readable and easier to understand. also specify the acronyms used using footnotes

What are the numbers in (). Means plus standard deviation??

Author Response

Reviewer 1.

Comments 1:

 I thank the authors for presenting this work. 
The topic addressed I believe is of substantial importance. Understanding the knowledge baseline of a production system, in this case that of a developing apese. defining the potential and limitations is fundamental to the evolution of an industry. 
However, the work needs revisions that will gliin its scientific value and comprehensibility and allow a better understanding of the results and state of the art generated.

Response 1:

Thank you for your comments. We have added the recommendations below (responses 2 to 5) as suggested

Comments 2:

Line 14: little is known about what?

Response  2:

The sentence  has been improved as follows (Page 1, lines 13 and 14)

Comments 3:

Introduction

I suggest that the introduction be improved by also reporting the breeds that are majorly bred in order to describe what is the livestock stock but also the production potential of these breeds.

Response 3:

 Information on breeds has been added (page 2, lines 56 and 57)

Comments 4:

Introduction

Please cite all references using reference numbers and place the numbers in square brackets (“[ ]”), e.g., [1], [1–3], or [1,3].

Response 4:

Has been addressed as suggested

Comments 5:

Methodology

It may be helpful to report the requested data and specify the mean of the questions. There are the range for example for milk productivity to define low or high?

Response 5:

Thank you for your comment, however, we have used the median value so a to also report the lowest and the highest scores

Comments 6:

Results

I suggest to add a descriptive statistics of the farms involved into the study (e.g, herd composition, numbers of animals for physiological category, milk yield, age of the farmers and level of edeucation…)

Response 6:

Thank you for your suggestion; however, detailed information can be found here: Ngou, A., Laven, R., Parkinson, T., Donaghy, D., & Kashoma, I. (2024). A survey of Smallholder dairy cattle farmers in Tanzania: Farmer Demographic Characteristics and Basic Management Constraints. Preprint. https://doi.org/10.21203/rs.3.rs-4993873/v2

Comments 7:

I suggest formatting the tables to make them more readable and easier to understand. also specify the acronyms used using footnotes

What are the numbers in (). Means plus standard deviation??

Response 7:

Addressed as suggested for all the tables

Reviewer 2 Report

Comments and Suggestions for Authors

General comments

Overall, I think this is a really nice manuscript that addresses a knowledge gap and for that reason has merit.  The work appears to be somewhat mixed methods, although the combination of qualitative data with quantitative analysis is a little confused and it is my opinion that the regression analysis of these data is unnecessary.  It is not well explained why data from different areas of Tanzania are compared using odds ratios and I do not think this approach really addresses your study question (i.e. to assess farmer knowledge of infertility) in a meaningful way.  Odds ratios are typically used to indicate the strength of association between a defined factor and the outcome - I struggle to understand why you are trying to show this here.  My suggestion would be to remove all the regression analysis and present the descriptive data only – this is the analysis that really illustrates the farmer knowledge and regional variations in a meaningful way and adds valuable insight.  If you do decide to continue to present the odds ratios and confidence intervals, I note that none of the odds ratios have p-values, so it is not possible for the reader to get an idea of how significant these results are - this is something that needs correcting. 

I would also like to see some discussion of the limitations of this study which is currently lacking from the discussion section.

Minor comments

Line 21: Consider starting this line with ‘our findings…’ for more active language and to be really clear you are referring to the findings of the present study (‘the findings…’ is a little vague). 

Line 26: The sentence ‘Despite its impact about how farmers understand and manage it’ doesn’t make sense – do you mean that there is little information available about how farmers understand and manage fertility?

Line 27: Knowledge does not need a capital letter here

Line 31: Rephrase to read ‘…from a list of ten signs…’ for improved clarity

Lines 42-43: I am curious as to why Tanzania is not one of your keywords given the focus of this study?

Line 60: Clarify that you are referring to Tanzania when you state smallholder farmers produce most milk (I think this is what you mean here?)

Line 75: This would be grammatically improved if it read ‘all of these effects reduce…’

Line 94: author does not need a capital here

Line 97: It would be beneficial to include some brief detail about how confidentiality was maintained and how the data were securely and confidentially stored. 

Lines 110-115: It is quite difficult to understand what you mean when describing your methods here.  I would consider re-writing this section to be clearer to readers who are not familiar with your study. 

Lines 126-142: Similarly, it is difficult to understand this description of your methods, consider re-writing to improve clarity. 

Line 143: Why was Tanga chosen as the reference region?

Line 152: Replace ‘is’ with ‘was’ (in the phrase ‘infertility is consistent’) to align with the conventional past tense format of presenting results

Lines 242-243: Here you report that you collapsed the five categories into three for analysis of regional differences, but the data are not presented in this way in the table which is a bit confusing.  I would either present the data in the table in the way you report analysing it, or offer an explanation as to why you have chosen to present the data in the table differently to how you report analysing it.

Lines 263-264: X-axis explanation should be in the figure caption, not the text.

Lines 269-280: I’m not sure that it is really necessary to include here whether the AICc increased or decreased when adding/removing factors - you should be presenting the results of the model with the best fit.  If you do include this information, a brief interpretation/explanation of this is warranted and it would also be helpful to state which was the final model you used. 

Figure and tables

General comments:

All graphs are missing x-axis and y-axis titles – please correct this.

I don’t really think it is necessary to explain interpretation of odds ratios beyond stating what the referent category is - the readers of this journal can be expected to be familiar with this.     

Specific line comments:

Figure 2: Correct spelling of ‘purulent’ on the y-axis

Figure 3: Clarify in the figure caption that these data are all regions combined (‘overall’ is a bit vague)

Author Response

Reviewer 2.

Comments 1:

 Overall, I think this is a really nice manuscript that addresses a knowledge gap and for that reason has merit.  The work appears to be somewhat mixed methods, although the combination of qualitative data with quantitative analysis is a little confused and it is my opinion that the regression analysis of these data is unnecessary.  It is not well explained why data from different areas of Tanzania are compared using odds ratios and I do not think this approach really addresses your study question (i.e. to assess farmer knowledge of infertility) in a meaningful way.  Odds ratios are typically used to indicate the strength of association between a defined factor and the outcome - I struggle to understand why you are trying to show this here.  My suggestion would be to remove all the regression analysis and present the descriptive data only – this is the analysis that really illustrates the farmer knowledge and regional variations in a meaningful way and adds valuable insight.  If you do decide to continue to present the odds ratios and confidence intervals, I note that none of the odds ratios have p-values, so it is not possible for the reader to get an idea of how significant these results are - this is something that needs correcting. 

Response 1:

Thank you, we have added description on how to interpret the OR, i.e. when the 95%CI exclude 1, indicates a P-Value of <0.05. Also we have used logistic regression in order to understand (in addition to frequences per category in the region) the differences that exists among the regions (how are they related to each other).

Comments 2.

I would also like to see some discussion of the limitations of this study which is currently lacking from the discussion section.

Response 2.

Addressed (page 24, lines 519 to 523)

Comments 3.

Line 21: Consider starting this line with ‘our findings…’ for more active language and to be really clear you are referring to the findings of the present study (‘the findings…’ is a little vague). 

Response 3.

Addressed as suggested (Line 20, page 1)

Comments 4..

Line 26: The sentence ‘Despite its impact about how farmers understand and manage it’ doesn’t make sense – do you mean that there is little information available about how farmers understand and manage fertility?

Response 4.

Thank you, yes, that was the meaning; however, we have put it as suggested.

Comments 5.

Line 27: Knowledge does not need a capital letter here

Response5.

Addressed as suggested (line 27)

Comments 6.

Line 31: Rephrase to read ‘…from a list of ten signs…’ for improved clarity

Response

Comments 7.

Lines 42-43: I am curious as to why Tanzania is not one of your keywords given the focus of this study?

Response 7.

Added Tanzania as a keyword (line 42)

Comments 8.

Line 60: Clarify that you are referring to Tanzania when you state smallholder farmers produce most milk (I think this is what you mean here?)

Response  8.

Addressed (line 62)

Comments 9.

Line 75: This would be grammatically improved if it read ‘all of these effects reduce…’

Response 9.

Addressed (line 78)

Comments 10.

Line 94: author does not need a capital here

Response  10

Addressed (line 96)

Comments 11.

Line 97: It would be beneficial to include some brief detail about how confidentiality was maintained and how the data were securely and confidentially stored. 

Response 11.

Improved as suggested (lines 98 to 100, on page 3)

Comments 12a.

  1. Lines 110-115: It is quite difficult to understand what you mean when describing your methods here.  I would consider re-writing this section to be clearer to readers who are not familiar with your study. 

Response 12a.

As mentioned in the first paragraph of the methodology section (lines 91 and 92), the detailed methodology could be found here: Ngou, A., Laven, R., Parkinson, T., Donaghy, D., & Kashoma, I. (2024). A survey of Smallholder dairy cattle farmers in Tanzania: Farmer Demographic Characteristics and Basic Management Constraints. Preprint. https://doi.org/https://doi.org/10.21203/rs.3.rs-4993873/v1. However, we have tried to improve for the reader to understand.

Comments  12b.

  1. Lines 126-142: Similarly, it is difficult to understand this description of your methods, consider re-writing to improve clarity. 

Response

As per number 12a above

Comments 13.

Line 143: Why was Tanga chosen as the reference region?

Response 13

Thank you for this observation. This was chosen by the software (after alphabetically arranging the regions). This allowed us to compare the regions

Comments 14.

Line 152: Replace ‘is’ with ‘was’ (in the phrase ‘infertility is consistent’) to align with the conventional past tense format of presenting results

Response 14.

Replacement was done (line 156)

Comments 15.

Lines 242-243: Here you report that you collapsed the five categories into three for analysis of regional differences, but the data are not presented in this way in the table which is a bit confusing.  I would either present the data in the table in the way you report analysing it, or offer an explanation as to why you have chosen to present the data in the table differently to how you report analysing it.

Response 15.

We appreciate your observation. We intended to allow the reader to see the full distribution of the responses by presenting the original five categories in the table. For analysis, we collapsed them into three categories because some regions had very few or zero responses in some categories, which made statistical comparison difficult.

Comments 16.

Lines 263-264: X-axis explanation should be in the figure caption, not the text.

Response 16.

Addressed (lines 247 to 248)

Comments 17.

Lines 269-280: I’m not sure that it is really necessary to include here whether the AICc increased or decreased when adding/removing factors - you should be presenting the results of the model with the best fit.  If you do include this information, a brief interpretation/explanation of this is warranted and it would also be helpful to state which was the final model you used. 

Response 17.

(WAITING FOR RICHARD TO ASSIST)

Figure and tables

Comments 18.

General

All graphs are missing x-axis and y-axis titles – please correct this.

Response 18.

Addressed as suggested

Comments 19.

I don’t really think it is necessary to explain interpretation of odds ratios beyond stating what the referent category is - the readers of this journal can be expected to be familiar with this.     

Response 19.

Thank you; however, we have done this to allow the reader to have a continuous flow while reading

Specific line comments:

Comments 20.

Figure 2: Correct spelling of ‘purulent’ on the y-axis

Response 20.

Thank you for your observation. Addressed

Comments 21.

Figure 3: Clarify in the figure caption that these data are all regions combined (‘overall’ is a bit vague)

Response 21.

Addressed by omitting  the word “overall” from line 246

Reviewer 3 Report

Comments and Suggestions for Authors

The paper titled "Knowledge, Attitudes, and Practices of Smallholder Dairy Cattle Farmers in Tanzania: A Cross-Sectional Survey on Cattle Infertility" addresses an important and timely topic. I find the subject matter of the article fascinating and read the manuscript with great interest. The paper aligns well with the scope of the journal. However I believe that in its current form it has several shortcomings.

Main Research Question

The primary question this research tackles is: What are the current knowledge attitudes and practices (KAP) of smallholder dairy cattle farmers in Tanzania regarding dairy cattle infertility? The study aim to fill a notable gap in understanding how these farmers perceive and manage fertility challenges ultimately aiming to inform targeted interventions for improving herd productivity.

Originality and Relevance to the Field

I absolutely think the topic is both original and highly relevant to the field of veterinary science and agricultural development. While studies on dairy-cattle-infertility exist, this one focuses specifically on smallholder farmers in Tanzania a context often overlooked in broader research. This addresse a significant gap because the challenges and practices of smallholder farmers in developing countries can differ vastly from large-scale commercial operations or those in more developed regions. Understand their specific KAP is crucial for developing effective, locally appropriate, and sustainable solutions, which is something that's really needed right now. Its provides a ground-up perspective which is often missing.

Contribution to the Subject Area

Compared to other published material, this paper offers a detailed, multi-regional survey of smallholder dairy farmer perspectives on infertility in Tanzania. Many existing studies might highlight the prevalence or causes of infertility from a clinical standpoint. However, this manuscript delves into the human element – what the farmers themselves know, how they feel about it, and what they actually do. The regional comparisons are particularly valuable, as they reveal variations in understanding and practices that can inform more nuanced policy and educational programs. For instance, the discussion on why farmers might intentionally delay calving in certain regions due to market conditions offers a practical insight not always covered in purely biological or economic papers. It really adds to the practical application of knowledge, you know?

Major Comments to the Author

The study is quite well-conceived and generally executed well, providing much-needed data on a critical topic. The detailed regional comparison definitely adds significant value, which is great. However, I've noted a few areas that could be improved upon to really enhance the paper's overall impact and scientific rigor.

Introduction

The introduction sets the stage quite nicely, clearly establishing the problem and the knowledge gap this study aims to fill. One minor, almost nitpicky, point: just ensure all references are consistently formatted throughout the entire manuscript, as noted in the [M1] comment on page 2. This is more of a style issue, but it can make a real difference in the final published version.

On line 48 I suggest include also 10.1007/s11250-025-04363-1.

Methodology

The sampling approach relying on convenience and snowball sampling is understandable given the context but does limit the generalizability of the findings to the broader Tanzanian smallholder dairy farmer population. A discussion acknowledging this limitation more prominently, perhaps within the methodology itself would strengthen the transparency of the research. Additionally it would be beneficial if the authors could include an appendix with the full questionnaire or at the very least, specify some of the exact questions posed to farmers especially concerning how "repeat breeding" was explained, as it deviates from the strict definition and could influence farmer responses. Finally while the discussion briefly touches on the potential for interviewer bias due to the first author's veterinary background expanding on this as a methodological limitation could further strengthen the paper.

Results

results are presented clearly and are quite comprehensive. However Figure 2, while informative, feels a little visually cluttered; the color choices for 'Correct', 'Incorrect', and 'Not sure' are quite similar, making it a bit difficult to distinguish at a quick glance. Perhaps a different color scheme would improve readability for the reader. Also while the interpretations of the odds ratios are generally sound, a bit more caution in discussion of those with very wide confidence intervals (e.g., lines 193-195) might be warranted as such wide intervals suggest less precise estimates, even if the point estimate itself is substantial. It's just good practice, I think.

Discussion

The discussion largely aligns with the evidence and arguments presented, and it effectively addresses the main research question. The insights regarding farmer education levels correlating with knowledge (lines 454-461) are compelling. Still, the direct link between the presented data (number of secondary schools, university location) and individual farmer education levels measured in this specific study isn't explicitly shown. If this correlation was directly assessed, clarifying that would be helpful; if not, framing it more as a plausible hypothesis for future research would be more accurate.

On lines 454-461, where the paper discusses the correlation between farmer education levels (like secondary schools and university locations) and knowledge, if this is being presented as a direct finding or a strong link, it needs a clear reference. If it's more of a hypothesis or general observation from outside the study's direct measurements, it should be framed as such, or external sources supporting this general correlation should be cited. It's a bit unclear in the current phrasing how this connection is derived within this paper's scope. I suggest read and cite: https://www.journals.uchicago.edu/doi/abs/10.1086/452139.

Similarly, the economic interpretations of market conditions influencing calving intervals (lines 481-499) are excellent and add significant depth, but it should be explicitly stated that these insights are based on external literature and market understanding rather than direct measurements from the survey itself. The discussion about underestimation of less frequent infertility causes, such as abortion or retained fetal membranes, is particularly insightful and well-supported, effectively bridging farmer perception with scientific importance.

The excellent insights on lines 481-499, which discuss how market conditions influence farmers' decisions on calving intervals, seem to draw on external economic understanding and literature. It's important to make sure that all the specific external sources that informed these interpretations are explicitly referenced. If there are broader economic studies or market analyses that underpin these points, they should definitely be included. I suggest https://doi.org/10.15232/pas.2015-01463 and https://doi.org/10.15567/mljekarstvo.2017.0107.

Conclusions

The conclusions effectively summarize the studys findings and underscore the critical need for targeted, region specific interventions and educational programs. They are consistent with the evidence provided and directly answer the main question posed by the research. The call for further qualitative studies to understand regional differences is a good suggestion.

Minor Comments to the Author

The manuscript is generally well-written, but a few small adjustments could enhance clarity and polish. These are mostly stylistic or very minor factual corrections.

Abstract

On line 25, there's a slight awkwardness in phrasing; perhaps rephrase "Despite its impact, about how farmers understand and manage it" for better flow. Something like "Despite its impact, little is understood about how farmers perceive and manage it." might work better.

Introduction

In the paragraph starting on line 59, the numbers like "-97%" and "-12 billion L/year" (lines 62 and 56) feel a bit informal. It would be better to replace the hyphen with a more formal symbol like a tilde (∼) or simply write "approximately" for a more academic tone.

Methodology

On line 114, "KoboToolbox (Cambridge, USA)" seems to have a small extra space before the closing parenthesis. Just a quick fix there. Also, regarding the statistical analysis on line 137, where it says "p<0.05", it might be helpful to ensure that this is consistently formatted throughout, perhaps with italicized 'p'. This is a very minor detail, but it makes the notation uniform.

Results

In Table 1, the headings "OR (95% CI)" appear for each region, but some cells within those columns are blank or have "***". It would be clearer to include a footnote or a key that explicitly states why these values are absent, e.g., "OR and 95% CI not calculable due to zero variance in responses." This would remove any ambiguity for the reader. Also, the term "Sillbirth" in Figure 2 and Table 1 (line 159) is likely a typo for "Stillbirth". Please correct this for consistency.

Discussion

On line 504, "Renata & VukoviĆ, 2013" has a special character in "VukoviĆ" that might not render correctly in all journal systems; it's generally safer to stick to standard Latin characters or ensure the journal's submission system supports such characters. Also, on line 512, "157.00 USD $)" has the dollar sign after the parentheses. It should probably be inside or the parentheses adjusted to include the dollar amount.

Author Response

Reviewer 3.

Comments 1:

The introduction sets the stage quite nicely, clearly establishing the problem and the knowledge gap this study aims to fill. One minor, almost nitpicky, point: just ensure all references are consistently formatted throughout the entire manuscript, as noted in the [M1] comment on page 2. This is more of a style issue, but it can make a real difference in the final published version.

Response 1.

Thank you for your suggestion. References have been made as advised

Comments 2.

The sampling approach relying on convenience and snowball sampling is understandable given the context but does limit the generalizability of the findings to the broader Tanzanian smallholder dairy farmer population. A discussion acknowledging this limitation more prominently, perhaps within the methodology itself would strengthen the transparency of the research. Additionally it would be beneficial if the authors could include an appendix with the full questionnaire or at the very least, specify some of the exact questions posed to farmers especially concerning how "repeat breeding" was explained, as it deviates from the strict definition and could influence farmer responses. Finally while the discussion briefly touches on the potential for interviewer bias due to the first author's veterinary background expanding on this as a methodological limitation could further strengthen the paper.

Response 2.

Addressed from lines 520 to 524 of the discussion

Comments 3.

results are presented clearly and are quite comprehensive. However Figure 2, while informative, feels a little visually cluttered; the color choices for 'Correct', 'Incorrect', and 'Not sure' are quite similar, making it a bit difficult to distinguish at a quick glance. Perhaps a different color scheme would improve readability for the reader. Also while the interpretations of the odds ratios are generally sound, a bit more caution in discussion of those with very wide confidence intervals (e.g., lines 193-195) might be warranted as such wide intervals suggest less precise estimates, even if the point estimate itself is substantial. It's just good practice, I think.

Response 3

Colours for Figure 2 have been changed as suggested.

Comments 4.

The discussion largely aligns with the evidence and arguments presented, and it effectively addresses the main research question. The insights regarding farmer education levels correlating with knowledge (lines 454-461) are compelling. Still, the direct link between the presented data (number of secondary schools, university location) and individual farmer education levels measured in this specific study isn't explicitly shown. If this correlation was directly assessed, clarifying that would be helpful; if not, framing it more as a plausible hypothesis for future research would be more accurate.

Response 4.

A statement was added from lines 447 to 450

Comment 5.

Similarly, the economic interpretations of market conditions influencing calving intervals (lines 481-499) are excellent and add significant depth, but it should be explicitly stated that these insights are based on external literature and market understanding rather than direct measurements from the survey itself. The discussion about underestimation of less frequent infertility causes, such as abortion or retained fetal membranes, is particularly insightful and well-supported, effectively bridging farmer perception with scientific importance.

Response 5

Thank you

Comment 6

The excellent insights on lines 481-499, which discuss how market conditions influence farmers' decisions on calving intervals, seem to draw on external economic understanding and literature. It's important to make sure that all the specific external sources that informed these interpretations are explicitly referenced. If there are broader economic studies or market analyses that underpin these points, they should definitely be included.

Response 6.

Addressed from lines 488 to 490

Comment 7.

Abstract

On line 25, there's a slight awkwardness in phrasing; perhaps rephrase "Despite its impact, about how farmers understand and manage it" for better flow. Something like "Despite its impact, little is understood about how farmers perceive and manage it." might work better.

Response 7.

Addressed (line 26)

Comment 8

In the paragraph starting on line 59, the numbers like "-97%" and "-12 billion L/year" (lines 62 and 56) feel a bit informal. It would be better to replace the hyphen with a more formal symbol like a tilde (∼) or simply write "approximately" for a more academic tone.

Response 9.

Addressed as suggested

Comment 10

On line 114, "KoboToolbox (Cambridge, USA)" seems to have a small extra space before the closing parenthesis. Just a quick fix there. Also, regarding the statistical analysis on line 137, where it says "p<0.05", it might be helpful to ensure that this is consistently formatted throughout, perhaps with italicized 'p'. This is a very minor detail, but it makes the notation uniform.

Response 10

Addressed as suggested

Comment 11

In Table 1, the headings "OR (95% CI)" appear for each region, but some cells within those columns are blank or have "***". It would be clearer to include a footnote or a key that explicitly states why these values are absent, e.g., "OR and 95% CI not calculable due to zero variance in responses." This would remove any ambiguity for the reader. Also, the term "Sillbirth" in Figure 2 and Table 1 (line 159) is likely a typo for "Stillbirth". Please correct this for consistency.

Response 11

Addressed (Figure 2 and Table 1)

Comment 12

On line 504, "Renata & VukoviĆ, 2013" has a special character in "VukoviĆ" that might not render correctly in all journal systems; it's generally safer to stick to standard Latin characters or ensure the journal's submission system supports such characters. Also, on line 512, "157.00 USD $)" has the dollar sign after the parentheses. It should probably be inside or the parentheses adjusted to include the dollar amount.

Response 12

Addressed as suggested

Round 2

Reviewer 1 Report

Comments and Suggestions for Authors

The authors have complied with the reviewers' requests. 

Reviewer 3 Report

Comments and Suggestions for Authors

all my comments were addressed properly

however, the paper is too specific to a geographical area of the world (Tanzania) and difficult to spread to an international stage. For this reason, i believe that this manuscript is not suitable for veterinary sciences, a high-level international journal